**∂ | Open Peer Review** | Clinical Microbiology | Research Article

# Comparison and validation of multiple machine learning algorithms for predicting MDRO infection in catheter-related bloodstream patients: a multicenter cohort study

Hongwei Wang,[1] Caizheng Yang,[2] Ming Zhao,[3] Fen Ren,[4] Xueyu Wang,[5] Haihua Yan,[1] Weiwei Qin,[1] Fangying Tian,[6] Linping Li[7]

**ABSTRACT** Early identification of patients at high risk for multidrug-resistant organism (MDRO) infection in catheter-related bloodstream infection (CRBSI) is crucial for precise antimicrobial therapy. This study aimed to develop and externally validate a machine learning (ML) model to predict this risk, thereby supporting early clinical intervention. Patients with CRBSI were extracted from the Medical Information Mart for Intensive Care IV database and classified into MDRO and non-MDRO groups based on microbiological culture and antimicrobial susceptibility testing. Missing data from 51 clinical variables were handled using Random Forest-based multiple imputation. Ten predictive features were selected by integrating correlation heatmap analysis, variance inflation factor, and least absolute shrinkage and selection operator regression. Eight ML models, including XGBoost and Random Forest, were constructed and tuned via hyperparameter optimization. The optimal model was selected primarily using the area under the receiver operating characteristic curve (AUC), supplemented by the F1-score, Brier score, accuracy, and recall. Its performance was further evaluated using a confusion matrix and calibration curve. External validation was performed on a real-world multi-center cohort ($n = 362$) to assess generalizability. Model interpretability was analyzed using SHapley Additive exPlanations (SHAP). A total of 1,251 patients with CRBSI were enrolled in the development cohort, among whom 189 (15.1%) were diagnosed with MDR-CRBSI. Significant differences were observed between the two groups in indicators of inflammatory status and organ functional reserve ($P < 0.05$). Ten predictive features were identified using least absolute shrinkage and selection operator (LASSO) regression. Among the models evaluated, XGBoost exhibited the best performance in the training set, with an AUC of 0.877 (95% CI: 0.854–0.900), and also demonstrated favorable results in other evaluation metrics. The model maintained robust predictive ability in the external multicenter validation cohort, achieving an AUC of 0.851 (95% CI: 0.826–0.876). SHAP analysis revealed that red blood cell distribution width (RDW), C-reactive protein (CRP), platelet count, pH, length of hospital stay, and class of antibiotics used as key predictors of MDR-CRBSI. Among the eight ML models developed and validated, XGBoost demonstrated superior performance in both internal and external validation. Its predictive capability is driven by 10 key variables, such as RDW and CRP, enabling early identification of high-risk MDR-CRBSI patients and providing a valuable tool for guiding precise antimicrobial therapy.

**IMPORTANCE** Catheter-related bloodstream infection (CRBSI) complicated by multidrug-resistant organism (MDRO) is associated with high mortality and treatment failure. The critical delay in conventional microbiological diagnosis often necessitates empirical broad-spectrum antibiotics, exacerbating antimicrobial resistance. Our study develops and validates an interpretable machine learning model using readily available clinical variables to accurately predict the risk of MDR-CRBSI at an early stage. This tool

**Peer Reviewer** Ahmad Ahmad, International Centre for Genetic Engineering and Biotechnology, New Delhi, Delhi, India

Address correspondence to Linping Li, sxmu1010@163.com, Caizheng Yang, yangcaizheng@sxtbu.edu.cn, or Fangying Tian, tfy8048@163.com.

The authors declare no conflict of interest.

addresses a pressing clinical need by enabling timely, targeted antimicrobial therapy, thereby potentially improving patient outcomes and supporting antimicrobial stewardship efforts in the global fight against resistance.

**KEYWORDS** catheter-related bloodstream infection, multidrug-resistant organism, machine learning, SHAP, prediction model

Catheter-related bloodstream infection (CRBSI) is defined as a bloodstream infection attributable to an intravascular catheter, confirmed microbiologically in the absence of another identifiable source, and commonly encountered in the use of central venous, peripheral, and dialysis catheters (1–3). These devices are widely employed in critically ill patients for intravenous medication, parenteral nutrition (2, 4), and hemodialysis (5), making CRBSI a major healthcare-associated infection that significantly prolongs hospitalization, increases costs, and raises mortality. In recent years, improved infection control awareness has shifted the epidemiological profile of CRBSI: in the intensive care unit (ICU), systematic interventions—such as hand hygiene, chlorhexidine skin antisepsis, full barrier precautions, and daily catheter assessment (6)—have markedly reduced central venous catheter (CVC)-related infections, whereas peripheral catheter-related CRBSI has been on the rise (7–9). Moreover, CRBSI incidence continues to increase in non-ICU settings, including general wards, dialysis centers, and outpatient clinics, underscoring the need to re-examine and broaden the current focus of CRBSI prevention (10). Notably, against the backdrop of broad-spectrum antibiotic overuse, the pathogen spectrum in CRBSI is shifting toward Gram-negative bacilli (GNB), which now account for approximately 20%–25% of CRBSI cases, nearly half of which involve multidrug-resistant (MDR) organisms (11, 12). The high mortality, adaptable resistance mechanisms, and limited therapeutic options associated with MDR Gram-negative CRBSI represent an emerging challenge in infection control. Thus, early recognition of MDR-CRBSI is essential to guiding targeted antimicrobial therapy and improving patient outcomes.

However, the early identification of MDR-CRBSI remains challenging. Conventional blood culture and antimicrobial susceptibility testing require 24–72 h, necessitating empirical antimicrobial therapy during this window, which not only exacerbates selection pressure for resistance but may also lead to delayed or suboptimal management due to false-negative results—partly attributable to inadequate bacterial detachment from biofilms (13). Furthermore, physician assessment is subjective and prone to inter-observer variability, particularly in patients with complex comorbidities, increasing the risk of misjudgment and overtreatment (14). Individual biomarkers also exhibit limited predictive performance; for instance, white blood cell (WBC) count has poor and variable discriminatory power for differentiating between Gram-positive and Gram-negative infections (15). Finally, risk factors for MDR-CRBSI are multifactorial and interact in complex ways, involving not only age and underlying diseases but also healthcare exposures—such as renal replacement therapy, duration of mechanical ventilation, and prior antibiotic or immunosuppressant use. Conventional logistic regression struggles to capture such high-order interactions and is vulnerable to multicollinearity when multiple variables are included, often resulting in poor generalizability of prediction models.

Given the escalating challenge of antimicrobial resistance, there is an urgent need to develop advanced, data-driven prediction strategies. Machine learning (ML), with its superior capacity for capturing complex, non-linear relationships in high-dimensional data, offers a promising direction for early MDR-CRBSI prediction. While traditional statistical models show limited discriminative ability in predicting outcomes such as bacterial colonization, ML approaches leveraging electronic health records have demonstrated excellent performance in stratifying MDR infection risk (area under the curve [AUC] = 0.79) among high-risk populations, such as hematological patients and have identified patient clusters with unsupervised learning where up to 75% of MDR infections are concentrated (16–18). Beyond clinical variable-based prediction, ML has

been integrated with matrix-assisted laser desorption/ionization time-of-flight mass spectrometry (MALDI-TOF MS), enabling rapid prediction of pathogen resistance directly from mass spectra within hours (AUC = 0.95) (19). Similarly, Keith and colleagues applied ML models based on bacterial genomic features to predict phage susceptibility, extending the application of ML from risk forecasting to precision bacteriophage therapy applied ML models based on bacterial genomic features to predict phage susceptibility, extending the application of ML from risk forecasting to precision bacteriophage therapy (20). Despite these advances in ML for severe infections, a dedicated ML-based model for predicting MDRO infections specifically in CRBSI remains lacking. Therefore, this study aims to develop and validate an ML model incorporating multidimensional clinical features to facilitate early identification and precise management of MDR-CRBSI.

## MATERIALS AND METHODS

### Study design

We employed a retrospective two-cohort design to develop and validate a ML model for predicting MDR-CRBSI. The study design roadmap is presented in Fig. 1a. The training cohort was derived from the electronic health records (2008–2019) of the Medical Information Mart for Intensive Care IV (MIMIC-IV) database (V3.1). This database contains de-identified clinical data from patients admitted to the ICU of Beth Israel Deaconess Medical Center in Boston, USA, comprising over 450,000 hospitalization records (21, 22). As an internationally recognized clinical database, MIMIC-IV is distinguished by its scientifically structured architecture. All data undergo rigorous de-identification and systematic quality control procedures. The variables are clearly defined and standardized, with uniform recording formats. This ensures superior scientific validity, completeness, and reliability compared with retrospectively collected data from independent single-center or multi-center studies, thereby providing a high-quality foundational data set for model development. The authors obtained approval for accessing the MIMIC-IV database in October 2024 (ID: 63970598). As all information in MIMIC-IV is anonymized, the requirement for informed consent was waived.

The external validation data set was sourced from three regional medical centers in China. The primary data originated from the Second Hospital of Shanxi Medical University (SXMUSH), supplemented by data from the Neurosurgical ICU of the Fifth Hospital of Shanxi Medical University (SXMUFH-NICU) and the Infectious Diseases Department of the General Hospital of Ningxia Medical University (NXMUGH-ID), covering the period from January 2018 to January 2025. As a key tertiary care center in Shanxi Province, SXMUSH maintains 2,700 inpatient beds and admits approximately 20,000 patients annually. It manages a substantial proportion of severe trauma and critically ill patients within the province, making its CRBSI data highly representative. The other two participating centers also possess strong clinical capabilities and broad patient coverage within their respective regions. The data they provided complemented the primary data set from various perspectives, thereby enhancing the comprehensiveness and reliability of the external validation cohort.

### Study population and data extraction

CRBSI was diagnosed according to the criteria established by the Infectious Diseases Society of America (IDSA) (23). Specifically, CRBSI was defined as a bloodstream infection (confirmed by positive blood cultures) occurring in a patient with an indwelling vascular catheter or within 48 h of its removal, with laboratory evidence confirming the catheter as the source. Furthermore, MDR-CRBSI was defined as a CRBSI episode where the identified pathogen met the criteria for a MDRO, specifically exhibiting resistance to three or more classes of antimicrobial agents. Minimum inhibitory concentration (MIC) was determined and interpreted based on standards provided by the Clinical and Laboratory Standards Institute (CLSI) (24, 25). Importantly, when calculating the number

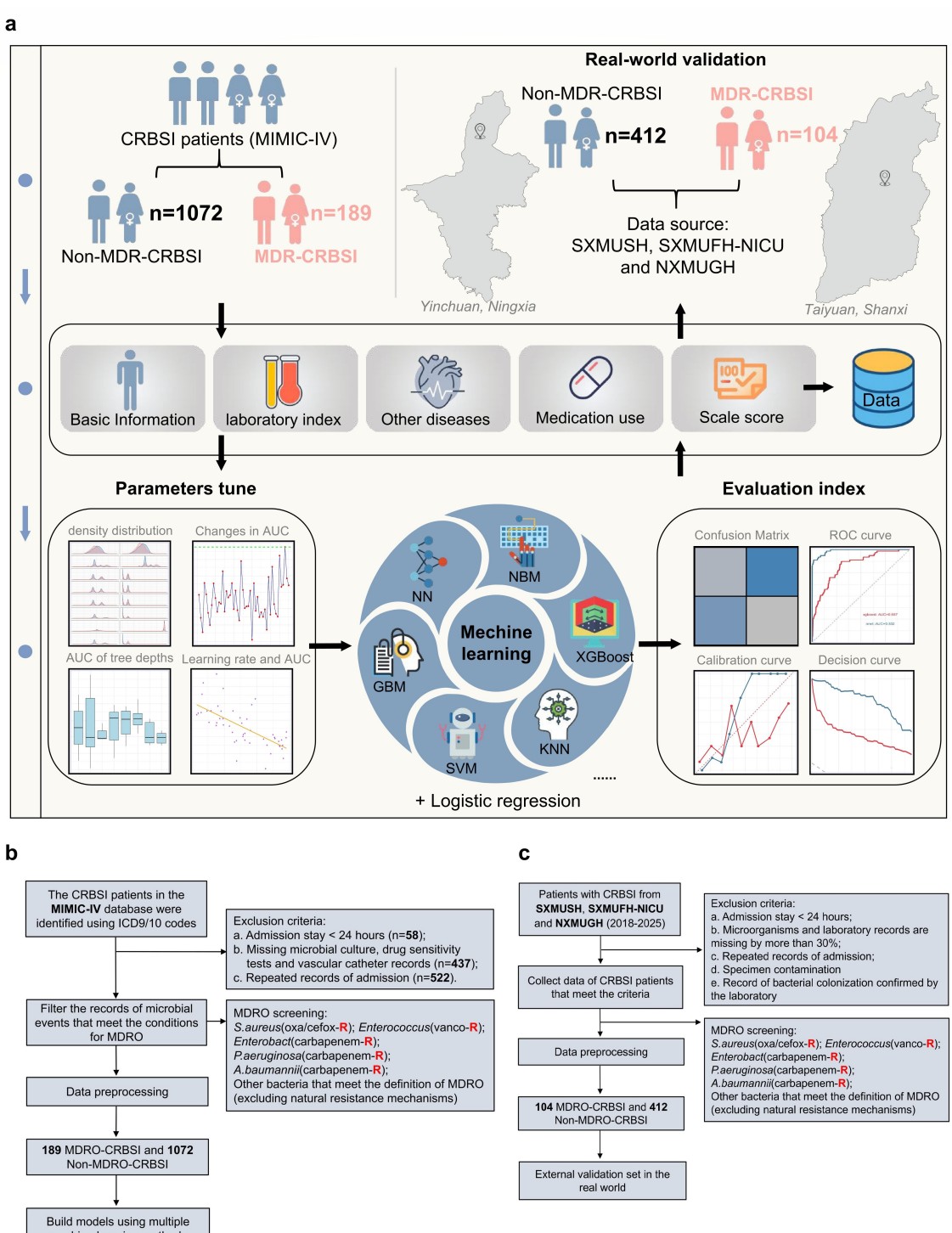

**FIG 1** Research summary chart and flowchart. (a) The model was developed on a training set from the MIMIC-IV database and externally validated on a multi-center cohort, utilizing logistic regression and seven ML algorithms with comprehensive performance metrics. (b) Training set flowchart: Patients with CRBSI were identified from the MIMIC-IV database by ICD codes, MDR status was determined by microbiological records, and exclusion criteria were set to obtain corresponding data for model construction. (c) An independent cohort of patients with CRBSI (2018–2025) from multiple centers was selected, and unified screening and exclusion criteria for multidrug-resistant organisms were applied to test the generalization ability of the model in real-world clinical scenarios.

of antimicrobial classes to which the bacterium was resistant, intrinsic resistance was not counted. For polymicrobial infections, patients were classified into the MDR-CRBSI group if at least one isolated pathogen fulfilled the MDRO definition. Following the common

practice of the U.S. Centers for Disease Control and Prevention (CDC), organisms producing extended-spectrum β-lactamases (ESBL) were not classified as MDROs in this study and were therefore excluded.

Patients with CRBSI were initially identified from the MIMIC-IV database using relevant International Classification of Diseases (ICD-9 and ICD-10) codes. From this initial cohort, we excluded patients with a hospital stay of less than 24 h, those missing records of microbiological cultures, antimicrobial susceptibility testing, or vascular catheter placement, as well as repeated, non-first hospital admissions. We specifically targeted five key MDRO types: methicillin-resistant *Staphylococcus aureus* (MRSA), vancomycin-resistant *enterococci* (VRE), carbapenem-resistant *Enterobacterales* (CRE), carbapenem-resistant *Pseudomonas aeruginosa* (CRPA), and carbapenem-resistant *Acinetobacter baumannii* (CRAB), alongside other microbiological events meeting our broader MDRO definition. Based on this, patients were stratified into two groups: the MDR-CRBSI group and the non-MDR-CRBSI group. The detailed flowchart for the derivation cohort is presented in Fig. 1b.

For the external validation set, data were collected from the infection control systems of three distinct medical centers. Patient data for those diagnosed with CRBSI at these centers were independently extracted by three experienced investigators (H.W., M.Z., and C.Y.) using standardized case report forms to ensure consistency. The exclusion criteria for this cohort comprised: duplicate records from the same patient; hospitalization lasting less than 24 h; specimens deemed contaminated or identified as colonizing flora based on laboratory assessment; and cases where microbiological or key laboratory data were missing for ≥30% of variables, or where a consensus between two independent reviewers could not be reached regarding the confirmation of CRBSI. The patient selection process for the external validation data set is summarized in Fig. 1c.

## Study variables and outcomes

The selection of study variables was guided by a review of existing literature on MDRO infections in patients with CRBSI, as well as the data availability within the MIMIC-IV database and the infection databases from the three participating hospitals (26–29). The collected variables comprised the following categories: (i) Demographic characteristics: patient identifier, sex, age, ethnicity, body mass index (BMI), and length of hospital stay (LOS). (ii) Laboratory parameters: hemoglobin (Hb), platelet count (PLT), red blood cell distribution width (RDW), WBC, lymphocyte percentage, neutrophil percentage, absolute lymphocyte count, absolute neutrophil count, eosinophil count, basophil count, chloride, total calcium, glucose, magnesium, potassium, sodium, D-dimer, prothrombin time (PT), international normalized ratio (INR), serum amylase, partial pressure of carbon dioxide ($PaCO_2$), pH, partial pressure of arterial oxygen ($PaO_2$), anion gap (AG), total bilirubin (TBIL), serum creatinine (Scr), serum albumin (Alb), C-reactive protein (CRP), and blood urea nitrogen (BUN). (iii) Comorbid conditions: the presence of hypertension, cancer, diabetes mellitus, heart failure, liver disease, renal disease, chronic obstructive pulmonary disease (COPD), and anemia. (iv) In-hospital invasive procedures and support: placement of a gastric tube, receipt of renal replacement therapy, mechanical ventilation, and the duration of vascular catheterization. (v) Illness severity scores: SOFA score, Simplified Acute Physiology Score II (SAPS II), and Acute Physiology Score III (APS III). (vi) Medication exposure: number of antibiotic classes used, administration of vasopressors, and use of immunosuppressive agents. The primary outcome of this study was the occurrence of MDRO infection among patients diagnosed with CRBSI.

## Data processing and quality control

Following initial screening, a total of 1,261 eligible patient records were identified. Ten records were subsequently excluded due to missing data exceeding 30% for the assessed variables. Among the 51 candidate variables, eight (WBC count, lymphocyte percentage, absolute neutrophil count, eosinophil count, chloride, D-dimer, serum amylase, and blood creatinine concentration) demonstrated a missing data rate surpassing

the predetermined threshold of 30%. Missing values in the remaining data set were addressed using multiple imputation via the Random Forest algorithm, implemented with the missForest package. The final analytical cohort thus comprised 1,251 patients characterized by 43 variables for subsequent model development.

To assess inter-variable relationships, we generated a correlation matrix heatmap (Fig. 2d) complemented by variance inflation factor (VIF) analysis (Table S1). A pronounced negative correlation was observed between absolute lymphocyte count and neutrophil percentage (correlation coefficient = −0.81), indicating substantial informational overlap. Given that neutrophil percentage is more routinely utilized in clinical practice for directly assessing inflammation related to bacterial infection, absolute lymphocyte count was excluded from further analysis. Similarly, a strong positive correlation (correlation coefficient = 0.73) was found between APS III and SOFA scores. As the SOFA score provides a more focused assessment of organ dysfunction—a hallmark of MDRO infections often leading to multi-organ damage—the APS III score was removed. Furthermore, PT and INR exhibited a near-perfect positive correlation (coefficient = 0.99), suggesting virtual redundancy. VIF analysis quantitatively confirmed severe multicollinearity for both PT (VIF = 76.75) and INR (VIF = 76.41), with values far exceeding the accepted threshold of 10. As both parameters are central to coagulation assessment, and INR is essentially a standardized derivative of PT, we retained PT for its more direct reflection of baseline coagulation status and excluded INR to enhance model stability and reliability.

After the removal of absolute lymphocyte count, INR, and the APS III score, categorical variables were converted into factors and subsequently represented using a dummy variable matrix. Continuous variables were standardized via Z-score transformation (mean = 0, standard deviation = 1). To mitigate class imbalance, the Synthetic Minority Over-sampling Technique (SMOTE) was applied to the training set. Specifically, 873 synthetic samples were generated for the minority class (MDR-CRBSI, $n$ = 189), resulting in a balanced training set with 1,062 samples per class (total $n$ = 2,124). This approach aimed to equilibrate the class distribution for model development.

## Feature selection

To identify key predictors associated with MDR-CRBSI infection, we performed a sequential feature selection process. Starting from an initial pool of 51 candidate variables, which was refined to 43 variables after data preprocessing and removal of redundant features (see Data processing and quality control), we applied LASSO regression followed by an elastic net model. Initially, LASSO regression (L1 regularization) was applied for preliminary variable screening. The optimal regularization parameter $\lambda$ was determined via fivefold cross-validation. Two key $\lambda$ values were considered: $\lambda\_min$, which corresponds to the value yielding the minimum mean squared error (MSE), and $\lambda\_1se$, the largest $\lambda$ value within one standard error of $\lambda\_min$, which provides a more parsimonious model (Fig. 2a). The dynamic shrinkage of variable coefficients with increasing regularization ($\log\lambda$) is illustrated in the regularization path plot (Fig. 2b).

Recognizing that LASSO can exhibit instability in variable selection when predictors are highly correlated, we further employed an elastic net model, which combines L1 and L2 regularization. The model was configured with a mixing parameter $\alpha$ of 0.3 to balance the properties of LASSO and ridge regression. A fivefold cross-validation was conducted with an initial search range for the regularization strength parameter s from 0 to 1. To enhance the sensitivity for retaining potentially relevant variables, the search range for s was subsequently narrowed to 0–0.2, and the search resolution was increased (20 grid points) for parameter fine-tuning. The final elastic net model was trained using the $\lambda\_min$ value.

This refined procedure resulted in the retention of ten variables with non-zero coefficients for final model development. Figure 2c presents the magnitude and direction of the coefficients for these final selected features.

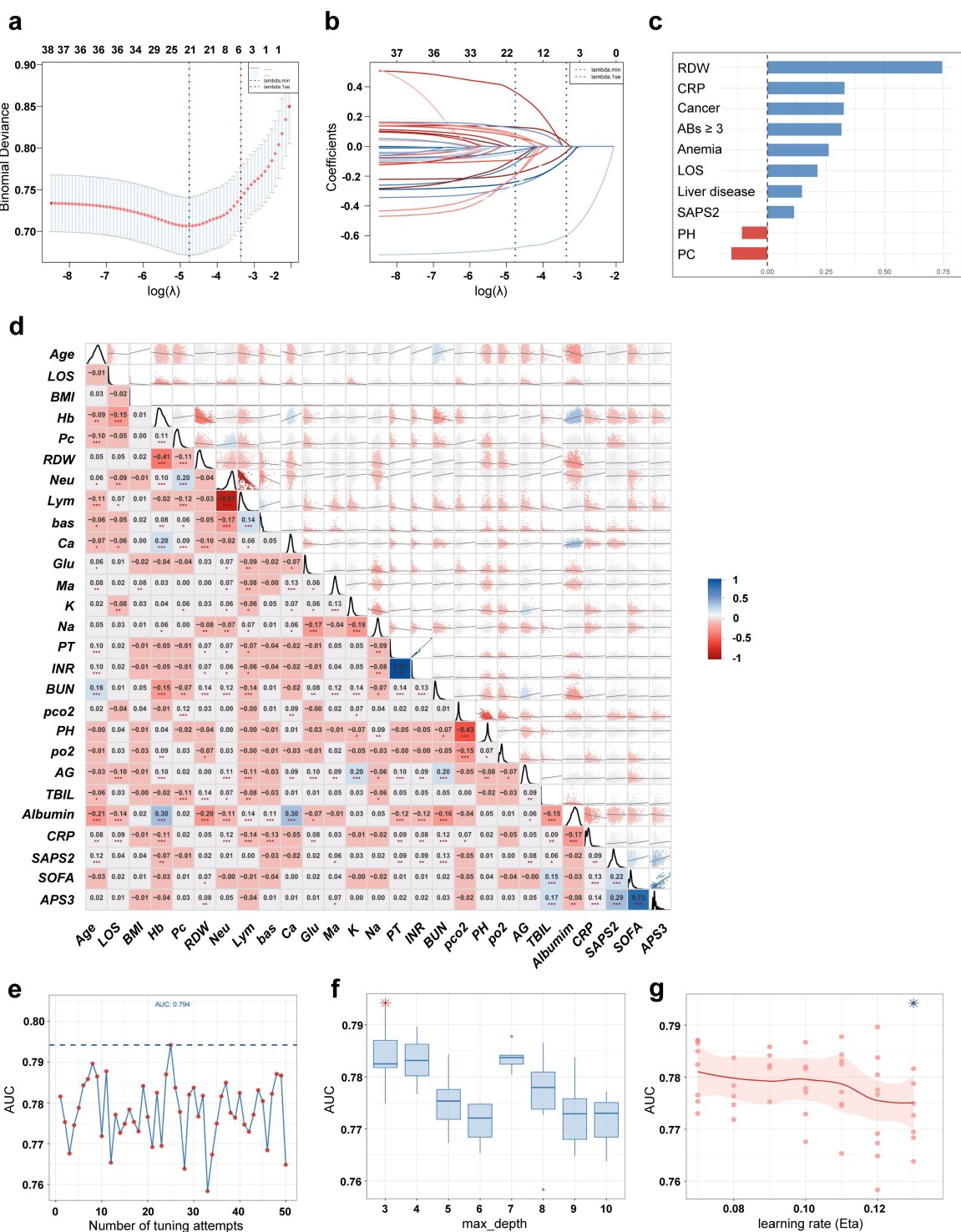

**FIG 2** LASSO regression, feature heatmap, and parameter optimization analysis. (a) LASSO regression path diagram. (b) LASSO regression cross-validation diagram. (c) Bar chart of feature importance. (d) Correlation heatmap of clinical characteristics. (e) Relationship diagram between AUC and number of parameter tuning attempts. (f) Box plot of relationship between AUC and tree depth (max_depth). (g) Relationship diagram between AUC and learning rate.

## Model development and parameter optimization

We subsequently developed prediction models using seven distinct ML algorithms—XGBoost, Neural Network (NN), K-Nearest Neighbors (KNN), Naïve Bayes, Support Vector Machine (SVM), Random Forest, and Gradient Boosting Machine (GBM)—alongside

conventional logistic regression. For each algorithm, hyperparameter tuning was conducted via a random search strategy coupled with 50 iterations of fivefold cross-validation. The optimal hyperparameter set for each model was identified based on the highest mean AUC derived from the cross-validation process. The final optimal model among all tuned candidates was selected by evaluating its comprehensive performance, primarily assessed by the AUC, with additional consideration of the F1-score, Brier score, accuracy, and recall.

To enhance the interpretability of the predictive model, we employed SHapley Additive exPlanations (SHAP). This method, rooted in cooperative game theory, quantifies the contribution of each input variable to individual predictions by fairly allocating the predictive output among all features (30). Throughout the analysis, identical data preprocessing steps were rigorously applied to both the training and external validation sets. The external validation was performed on a temporally independent sample, providing a comprehensive assessment of the model's generalizability across multiple dimensions, including discriminative ability, calibration, and overall classification performance.

## Statistical analysis

Data extraction from the MIMIC-IV database was performed using PostgreSQL 17. Subsequent data processing, statistical analysis, and ML modeling were carried out with SPSS 26.0 and R Studio (version 4.3.1). Descriptive statistics were applied based on data distribution: continuous variables conforming to a normal distribution are presented as mean (standard deviation), while non-normally distributed variables are summarized as median (interquartile range, IQR). Categorical variables are reported as counts (percentages). The normality of continuous variables was assessed using the Shapiro-Wilk test. Group comparisons were made using the $t$-test for normally distributed variables, Wilcoxon rank-sum test for non-normal distributions, and χ test or Fisher's exact test for categorical variables, as appropriate. A two-sided $P$-value of < 0.05 was considered statistically significant.

Addressing the critical challenge of standardization and reproducibility in ML workflows (31), we implemented a standardized analytical pipeline using the mlr3 framework in R. This framework was extended to incorporate custom preprocessing steps, including missing value imputation via the missForest package, and tailored hyperparameter tuning procedures. This approach ensured a consistent and reproducible model development process while accommodating the specific analytical requirements of this study (32).

## RESULTS

### Baseline characteristics

A total of 1,251 patients with CRBSI were enrolled in the training cohort, of whom 189 (15.1%) were diagnosed with MDR-CRBSI. No significant differences were observed between the two groups regarding demographic characteristics (age, sex, and BMI) ($P$ > 0.05). However, patients in the MDR-CRBSI group had a significantly longer mean hospital length of stay (9.46 days vs 6.67 days). Comparison of laboratory findings revealed that MDR-CRBSI patients exhibited a systemic inflammatory state and poorer organ functional reserve. Specifically, this was characterized by significantly lower Hb and PLT, alongside elevated levels of an inflammatory marker (CRP), a marker of hematopoietic stress (RDW), markers of organ dysfunction (BUN and TBil), and lower Alb levels. Regarding comorbidities, the MDR-CRBSI group carried a heavier burden of underlying diseases, with higher prevalence rates of cancer, heart failure, liver disease, and anemia. Consistent with the above findings, disease severity scores also indicated more critical illness in the MDR-CRBSI group. Furthermore, prior exposure to specific antibiotic classes was significantly associated with the MDR-CRBSI phenotype. The complete list of variables is provided in Table 1. The external validation data set

comprised three independent cohorts from different medical centers: the SXMUSH cohort (*n* = 182), SXMUFH-NICU cohort (*n* = 120), and NXMUGH-ID cohort (*n* = 60). The baseline information and descriptive statistics for continuous features in the external validation data set are shown in Table 2.

## Model development and parameter optimization

This study employed seven ML algorithms alongside logistic regression to construct prediction models, including XGBoost, RF, SVM, NN, KNN, NB, and LightGBM. All subsequent models were developed using the final set of 10 predictor variables selected through the feature selection process described above. All models underwent hyperparameter optimization via 50 rounds of random search combined with fivefold cross-validation to ensure optimal performance. Figure 2e illustrates the fluctuation trend of model AUC values across the 50 hyperparameter combination attempts. The blue dashed line indicates the highest AUC value of 0.794 achieved during the entire tuning process. Figure 2f displays the distribution of AUC values corresponding to different maximum depths of decision trees (max_depth). The light blue box plots represent the AUC distribution for each tree depth (the central line indicates the median, and the upper and lower edges represent the interquartile range). The box plot distribution shows that lower tree depths were associated with higher median AUC values, while excessively deep trees tended to cause performance degradation. Figure 2g shows the correlation between the learning rate (eta) hyperparameter and the AUC. Dots correspond to AUC values for different learning rates. The trend in the figure indicates that a learning rate around 0.13 yielded higher AUC values, whereas overly small learning rates might lead to suboptimal model performance.

Based on the optimization described above, we comprehensively evaluated all models on the training set and independent external validation sets. The optimal hyperparameter sets for all eight models, identified through this process, are detailed in Table S4. Figure 3a presents the ROC curves and corresponding AUC values for each model on the training set. Among them, the XGBoost model achieved the optimal AUC value (0.877 [95% CI: 0.854–0.900]). On the external validation sets (Fig. 3b), the XGBoost model maintained high performance, with an AUC of 0.851 (95% CI: 0.826–0.876). Multi-metric line charts (Fig. 3c and d) demonstrated that the XGBoost model also exhibited superior performance in terms of F1-score, Brier score, Accuracy, and Recall. The comprehensive multi-index data (Accuracy, Recall, F1-score, Brier Score) for all eight models in the training and validation sets are provided in Table S2 and Table S3, respectively. Furthermore, analysis of the confusion matrices on both the training and validation sets (Fig. 3e and f) revealed: In the training set, true positives (TPs) = 133, true negatives (TNs) = 1,057, false positives (FPs) = 5, and false negatives (FNs) = 56; in the validation set, TP = 52, TN = 286, FP = 2, and FN = 22. The proportions of FPs and FNs were low in both data sets, indicating that the model made few errors in identifying MDR-CRBSI, particularly showing high consistency in classifying the core target. Overall, the XGBoost model demonstrated excellent and stable performance across multiple evaluation dimensions.

## Model interpretation

To gain deeper insights into the model's decision-making mechanism, we performed SHAP analysis at both global and individual levels. Key predictors included RDW, CRP, PLT, PH, ABs, Los, SAPS2 score, anemia, liver disease, and cancer. We generated beeswarm plots for the training and external validation sets (Fig. 4a and e), which display the distribution of SHAP values for each feature across all samples. The results indicated that RDW and CRP were the most influential features for predicting MDR-CRBSI in both data sets: higher RDW values corresponded to more positive SHAP values, indicating a stronger positive contribution to MDR risk; similarly, higher CRP values were associated with significantly positive SHAP values, correlating positively with MDR risk. Feature dependence plots (Fig. 4b and f) illustrate the non-linear relationships between the

**TABLE 1** Characteristics of MDR-CRBSI and non-MDR-CRBSI patients of the MIMIC-IV database[a]

| Type | Feature | MDR-CRBSI ($n = 189$) | Non-MDR-CRBSI ($n = 1,062$) | Statistic | P |
|---|---|---|---|---|---|
| Basic | Age, year, M ($Q_1$, $Q_3$) | 59 (47, 67) | 58 (46, 69) | 0.164 | 0.87 |
| | Male, $n$ (%) | 100 (52.91) | 482 (45.39) | 3.651 | 0.056 |
| | BMI, kg/m², M ($Q_1$, $Q_3$) | 27.6 (24.2, 32.3) | 28.4 (24.1, 34.4) | 1.005 | 0.315 |
| | LOS, day, M (Q1, Q3) | 9.46 (2.87, 31.16) | 6.67 (2.89, 16.63) | −3.435 | 0.001 |
| Blood routine | Hb, g/dL, M ($Q_1$, $Q_3$) | 8.5 (7.6, 10.1) | 9.9 (8.4, 11.9) | 7.197 | <0.001 |
| | PLT, 10^9/L, M ($Q_1$, $Q_3$) | 129 (71.25, 239) | 211 (131, 297) | 5.768 | <0.001 |
| | Rdw, %, M ($Q_1$, $Q_3$) | 18.1 (16, 23) | 15.5 (14.1, 17.4) | −10.671 | <0.001 |
| | Neu, %, M ($Q_1$, $Q_3$) | 75.05 (61.05, 82.97) | 74.1 (61.1, 83.05) | 0.601 | 0.548 |
| | Bas, 10^9/L, M ($Q_1$, $Q_3$) | 0.2 (0, 0.5) | 0.3 (0.1, 0.6) | 3.960 | <0.001 |
| Blood biochemistry | Ca, mg/dL, M ($Q_1$, $Q_3$) | 8.4 (7.9, 9) | 8.6 (8.1, 9.1) | 2.474 | 0.013 |
| | Glu, mg/dL, M ($Q_1$, $Q_3$) | 125 (102, 172) | 120 (100, 156) | −2.032 | 0.042 |
| | Ma, mmol/L, M ($Q_1$, $Q_3$) | 2 (1.7, 2.2) | 2 (1.8, 2.2) | 1.260 | 0.208 |
| | K, mmol/L, M ($Q_1$, $Q_3$) | 4 (3.7, 4.5) | 4.1 (3.7, 4.5) | 0.272 | 0.786 |
| | Na, mmol/L, M ($Q_1$, $Q_3$) | 138 (135, 141) | 138 (135, 141) | 0.265 | 0.791 |
| | BUN, mg/dL, M ($Q_1$, $Q_3$) | 27 (18, 42) | 21 (12, 36) | −4.761 | <0.001 |
| | TBil, mg/dL, M ($Q_1$, $Q_3$) | 0.6 (0.3, 1.6) | 0.4 (0.3, 0.8) | −4.228 | <0.001 |
| | Alb, g/dL, M ($Q_1$, $Q_3$) | 3.2 (2.8, 3.8) | 3.5 (2.9, 4.1) | 4.485 | <0.001 |
| | CRP, mg/L, M ($Q_1$, $Q_3$) | 55.1 (31.7, 102.7) | 22.25 (4.2, 76.35) | −6.053 | <0.001 |
| | PT, s, M ($Q_1$, $Q_3$) | 14.05 (12.5, 16.33) | 13.5 (12.1, 15.9) | −2.308 | 0.021 |
| Blood gas analysis | PaCO₂, mmol/L, M ($Q_1$, $Q_3$) | 40 (34.5, 47) | 40 (35, 46) | 0.327 | 0.744 |
| | PH, M ($Q_1$, $Q_3$) | 7.38 (7.32, 7.44) | 7.39 (7.33, 7.43) | 1.319 | 0.187 |
| | PaO₂, mmol/L, M ($Q_1$, $Q_3$) | 89 (58.5, 128.5) | 90 (59, 130) | 0.471 | 0.638 |
| | AG, mmol/L, M ($Q_1$, $Q_3$) | 13 (11, 17) | 14 (12, 17) | 1.497 | 0.134 |
| Comorbidities | Hypertension, $n$ (%) | 132 (69.84) | 740 (69.68) | 0.002 | 0.965 |
| | Cancer, $n$ (%) | 76 (40.21%) | 297 (27.97) | 11.497 | 0.001 |
| | Diabetes, $n$ (%) | 77 (40.74) | 430 (40.49) | 0.004 | 0.948 |
| | Heart_failure, $n$ (%) | 84 (44.44) | 382 (35.97) | 4.930 | 0.026 |
| | Liver_disease, $n$ (%) | 74 (39.15) | 252 (23.73) | 19.811 | <0.001 |
| | Renal_disease, $n$ (%) | 77 (40.74) | 394 (37.1) | 0.906 | 0.341 |
| | COPD, $n$ (%) | 57 (30.16) | 283 (26.65) | 0.999 | 0.317 |
| | Anemia, $n$ (%) | 134 (70.9) | 645 (60.73) | 7.056 | 0.008 |
| Severity scores | SAPS2, M ($Q_1$, $Q_3$) | 36 (27, 49) | 34 (25, 44) | −1.987 | 0.047 |
| | SOFA, M ($Q_1$, $Q_3$) | 3 (2,6) | 2 (1,4) | −4.927 | <0.001 |
| Treatment | Antibiotic, $n$ (%) | | | 17.453 | 0.001 |
| | 0 | 34 (17.99) | 171 (16.10) | | |
| | 1 | 79 (41.80) | 594 (55.93) | | |
| | 2 | 57 (30.16) | 246 (23.16) | | |
| | 3 | 19 (10.05) | 51 (4.80) | | |
| | IMM, $n$ (%) | 34 (17.99) | 148 (13.94) | 2.121 | 0.145 |

[a]Continuous values were presented as median (interquartile range). Categorical values were presented as number (percentage). LOS: length of hospital stay; Hb: hemoglobin; PLT: platelet count; RDW: red blood cell distribution width; Neu: neutrophil percentage; Bas: basophil count; BUN: blood urea nitrogen; TBil: total bilirubin; Alb: serum albumin; CRP: C-reactive protein; PT: prothrombin time; PaCO₂: partial pressure of carbon dioxide; PaO₂: partial pressure of arterial oxygen; AG: anion gap.

values of continuous features and their corresponding SHAP values via scatter plots with trend lines, revealing the dynamic patterns of feature impact on predictions. These figures contain scatter plots for four features: RDW, PLT, CRP, and PH, with the horizontal axis representing the feature value and the vertical axis representing the SHAP value. Taking RDW as an example, the results show that higher RDW values correspond to significantly positive SHAP values. Finally, for specific individual samples, force plots (Fig. 4c and d) demonstrate how the prediction for a single sample is derived from the cumulative sum of the SHAP values of its features. E[f(x)] is the base value (mean model prediction), f(x) is the final predicted value for that sample, and each feature's ""contribution block"" represents its SHAP value (yellow pushes the prediction towards higher risk

**TABLE 2** Descriptive statistics of basic information and continuous feature of external validation data set

| Feature | Center A ($n$ = 182) | Center B ($n$ = 120) | Center C ($n$ = 60) |
|---|---|---|---|
| Age, year, M ($Q_1$, $Q_3$) | 51 (41.25, 60) | 50 (45, 55.25) | 50 (37.5, 58) |
| Male, $n$ (%) | 111 (60.99) | 73 (60.83) | 50 (83.33) |
| BMI, kg/m$^2$, M ($Q_1$, $Q_3$) | 24.53 ± 2.93 | 24.4 ± 3.01 | 24.1 ± 2.98 |
| LOS, day, M (Q1, Q3) | 14 (12, 16) | 14 (14, 17) | 15 (14, 21) |
| Hb, g/dL, M ($Q_1$, $Q_3$) | 11.55 (10.6, 12.29) | 11.7 (10.5, 12.41) | 10.16 (9.35, 11.04) |
| PLT, 10^9/L, M ($Q_1$, $Q_3$) | 178 (146.25, 237.75) | 167.5 (91, 232.25) | 216.5 (163, 253.75) |
| Rdw, %, M ($Q_1$, $Q_3$) | 16 (15, 18) | 16 (15, 18) | 16 (14, 18) |
| Neu, %, M ($Q_1$, $Q_3$) | 57.2 (49.4, 64.85) | 58.8 (50.17, 67.2) | 59.4 (49.5, 75.9) |
| Bas, 10^9/L, M ($Q_1$, $Q_3$) | 0.36 (0.19, 0.44) | 0.37 (0.23, 0.45) | 0.38 (0.22, 0.45) |
| Ca, mg/dL, M ($Q_1$, $Q_3$) | 9.11 (8.29, 9.66) | 9.11 (8.55, 9.62) | 10.23 (9.36, 10.87) |
| Glu, mg/dL, M ($Q_1$, $Q_3$) | 117 (107, 134) | 126.5 (113.75, 140.25) | 126 (113.5, 167) |
| Ma, mmol/L, M ($Q_1$, $Q_3$) | 1.59 (1.19, 2.47) | 1.64 (1.14, 2.46) | 1.61 (1.1, 2.22) |
| K, mmol/L, M ($Q_1$, $Q_3$) | 5.72 (5.1, 7.48) | 5.06 (4.64, 5.57) | 5.16 (4.64, 6.53) |
| Na, mmol/L, M ($Q_1$, $Q_3$) | 135 (127.25, 143) | 137.5 (131, 144) | 134.5 (127, 142) |
| BUN, mg/dL, M ($Q_1$, $Q_3$) | 16.51 (15.39, 18) | 15.79 (12.14, 17.2) | 12.44 (11.44,1 6.66) |
| TBil, mg/dL, M ($Q_1$, $Q_3$) | 2.1 (1.3, 3.38) | 2.1 (1.3, 4.2) | 12.95 (5.85, 17.9) |
| Alb, g/dL, M ($Q_1$, $Q_3$) | 2.37 (1.71, 3.21) | 2.21 (1.66, 2.9) | 2.15 (1.73, 2.86) |
| CRP, mg/L, M ($Q_1$, $Q_3$) | 13.98 (10.3, 19.09) | 12.28 (6.89, 17.82) | 10.35 (6.42, 18) |
| PT, s, M ($Q_1$, $Q_3$) | 11.3 (9.79, 12.2) | 11.5 (9.77, 12.3) | 11.75 (11.2, 12.72) |
| PaCO$_2$, mmol/L, M ($Q_1$, $Q_3$) | 34 (27, 48) | 42 (28.75, 66) | 40 (24.75, 71.25) |
| PH, M ($Q_1$, $Q_3$) | 6.72 (6.41, 6.98) | 6.71 (6.45, 6.96) | 7.38 (7.04, 7.66) |
| PaO$_2$, mmol/L, M ($Q_1$, $Q_3$) | 175 (38, 265.75) | 176.5 (38.75, 259.75) | 47.5 (31.75, 258) |
| AG, mmol/L, M ($Q_1$, $Q_3$) | 22 (16, 27.85) | 22 (16, 27) | 25 (22, 31.25) |
| Hypertension, $n$ (%) | 54 (29.67) | 29 (24.17) | 25 (41.67) |
| Cancer, $n$ (%) | 24 (13.19) | 33 (27.5) | 6 (10%) |
| Diabetes, $n$ (%) | 158 (86.81) | 96 (80) | 41 (68.33) |
| Heart_failure, $n$ (%) | 38 (20.88) | 30 (25) | 22 (36.67) |
| Liver_disease, $n$ (%) | 74 (40.66) | 38 (31.67) | 34 (56.67) |
| Renal_disease, $n$ (%) | 105 (57.69) | 50 (41.67) | 26 (43.33) |
| COPD, $n$ (%) | 20 (10.99) | 9 (7.5) | 5 (8.33) |
| Anemia, $n$ (%) | 16 (8.79) | 12 (10) | 15 (25) |
| SAPS2, M (Q1, Q3) | 31 (14.5, 38) | 32 (15.75, 38.25) | 31.5 (0, 38.25) |
| SOFA, M (Q1, Q3) | 2 (1, 4) | 3 (2, 6) | N.A[a] |
| Antibiotic, $n$ (%) | | | |
| 0 | 41 (22.53) | 26 (21.67) | 7 (11.67) |
| 1 | 87 (47.80) | 46 (38.33) | 27 (45) |
| 2 | 39 (21.43) | 43 (35.83) | 23 (38.33) |
| 3 | 14 (7.69) | 11 (9.17) | 3 (5) |
| VAS, $n$ (%) | 19 (10.44) | 14 (11.67) | 5 (8.33) |
| IMM, $n$ (%) | 106 (58.24) | 44 (36.67) | 12 (20) |

[a] N.A, not available (missing data).

of MDR-CRBSI, purple towards lower risk). For instance, in the example shown, PH = 0.869 contributes +0.0106, CRP = 2.64 contributes +0.0492; these feature values push the model's prediction towards classifying "this sample as MDR-CRBSI." In summary, the SHAP analyses consistently indicated that the association patterns between the values of the core included features and MDR-CRBSI risk were highly consistent across the training and external validation sets, suggesting that the feature interpretation of the optimal model is both reliable and generalizable.

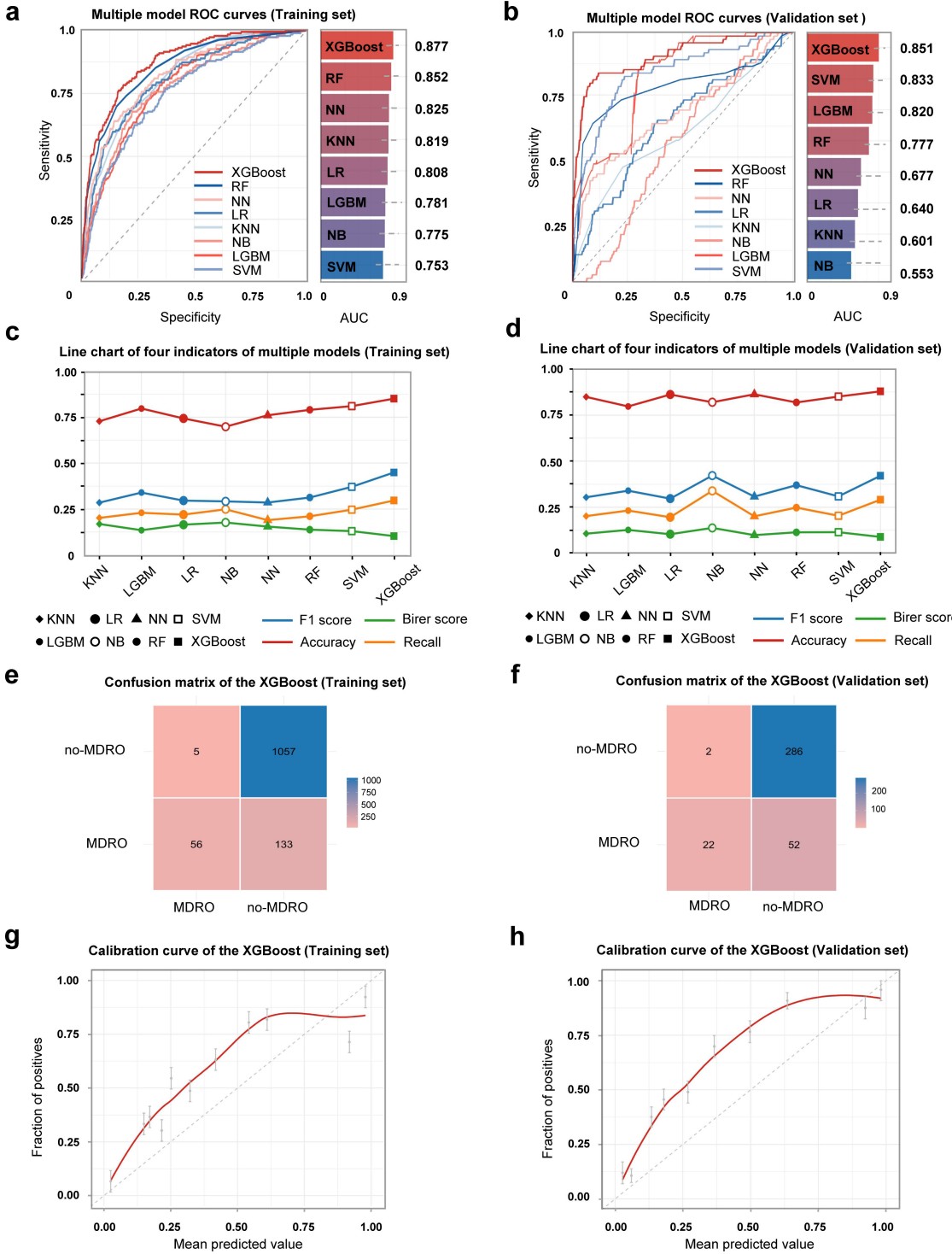

**FIG 3** Accuracy analysis of ML model. (a) Multiple model ROC curves (Training set). (b) Multiple model ROC curves (Validation set). (c) Line chart of four indicators of multiple models (Training set). (d) Line chart of four indicators of multiple models (Validation set). (e) Confusion matrix of the XGBoost (Training set). (f) Confusion matrix of the XGBoost (Validation set). (g) Calibration curve of the XGBoost (Training set). (h) Calibration curve of the XGBoost (Validation set).

## DISCUSSION

CRBSI is a common HAI, characterized by bacterial colonization of the catheter biofilm leading to systemic dissemination. When complicated by MDRO infection, CRBSI can significantly exacerbate damage to multiple systems, including the circulatory,

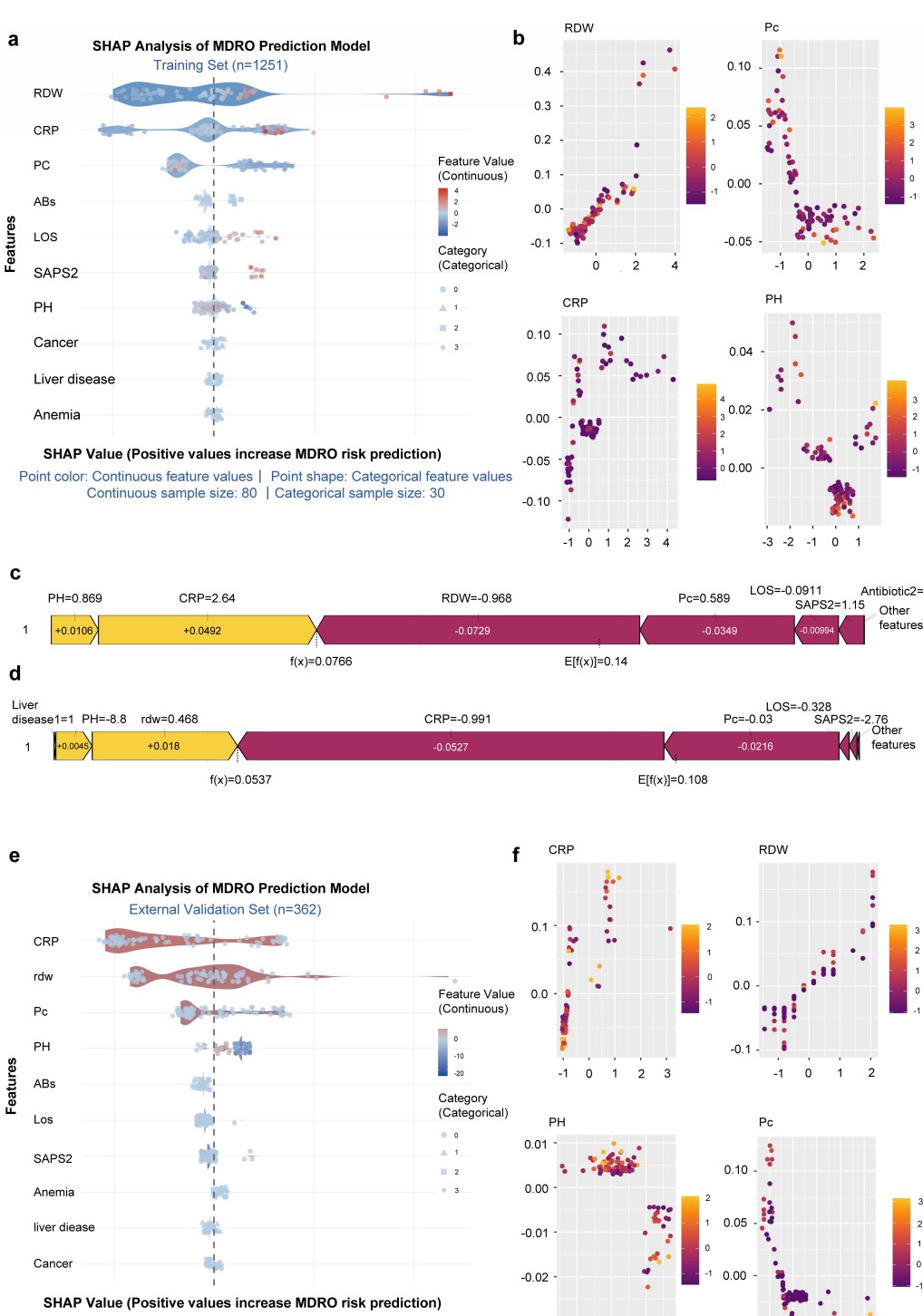

**FIG 4** Calculation results and SHAP interpretation of ML model for MDR-CRBSI. (a) The SHAP explanation swarm graph of the ML in the training set (showing the distribution of the contribution of each feature to the model's prediction). (b) Scatter plot of SHAP values among features (Training set). (c) SHAP explanation of a single sample in the ML mode (Training set). (d) SHAP explanation of a single sample in the ML mode (Validation set). (e) The SHAP explanation swarm graph of the ML in the validation set. (f) Scatter plot of SHAP values among features (Validation set).

respiratory, and urinary systems, potentially progressing to sepsis, shock, multiple organ failure, and even death (33). A pivotal challenge in clinical management stems from the

inherent delay associated with culture-based diagnostics, which creates a therapeutic gap often bridged by empirical broad-spectrum antibiotic therapy. While this well-intentioned but non-targeted approach addresses immediate life-threatening infection, it inadvertently perpetuates the cycle of antimicrobial resistance by exerting strong selective pressure on microbial populations (34). To bridge this critical gap between diagnosis and therapeutic intervention, our study developed and validated a prediction model integrating ten key clinical variables encompassing hematological indices, inflammatory markers, physiological parameters, comorbidities, and treatment-related factors. The model demonstrated robust performance across multiple clinical centers. Coupled with SHAP-based interpretability, it holds potential as a clinically applicable tool for early risk stratification, thereby creating opportunities for more precise and responsible antimicrobial stewardship.

The pathophysiology linking CRBSI and sepsis is key to understanding MDRO infection risk. Localized catheter-related infection, if not controlled promptly, can progress to sepsis via "biofilm shedding-bacteremia-systemic inflammatory activation." Conversely, the immune dysregulation and organ dysfunction induced by sepsis significantly increase the risk of MDRO colonization and infection, establishing a vicious cycle (35). Previous studies have validated the predictive value of RDW, PLT, and CRP for MDRO, aligning with our findings. Cui et al., in predicting MDRO infection in patients with VAP, identified elevated RDW as the most critical determinant of MDRO-VAP risk (29). Similarly, Li et al. included RDW as a core laboratory indicator in a model predicting MDRO colonization in the ICU, suggesting its utility across different severe infection scenarios (36). Our study, set in the context of CRBSI, further corroborates the value of RDW, which reflects severe inflammation-driven dysregulation of erythropoiesis and resulting erythrocyte heterogeneity. Furthermore, this hematological dysregulation coincides with thrombocytopenia, indicating a critical intersection between coagulopathy and immune dysfunction—a hallmark of severe sepsis where PLT consumption occurs alongside impaired microbial clearance (37). Additionally, CRP, a classical inflammatory marker, not only indicates the intensity of systemic inflammation but its persistent elevation often drives the use of broad-spectrum antibiotics, thereby exacerbating selective pressure for resistance (38). Beyond these systemic markers, our model also captures significant physiological derangement through parameters like decreased pH, where acidosis resulting from tissue hypoperfusion creates an environment that further compromises neutrophil function and may enhance biofilm stability for certain pathogens.

The inclusion of specific comorbidities (liver disease, cancer, anemia) underscores the importance of compromised host defenses in MDRO infection. Liver dysfunction impairs humoral immunity via reduced complement production and promotes bacterial translocation through disruption of the gut-liver axis (39, 40). Malignancies, particularly hematological cancers, induce profound immunosuppression, often exacerbated by cytotoxic therapies (41). This immunocompromised state, compounded by frequent healthcare exposures such as recurrent hospitalizations, creates a high-risk milieu for MDRO acquisition and infection. Similarly, anemia often reflects underlying states of chronic inflammation, nutritional deficiency, or bone marrow suppression, leading to reduced tissue oxygenation and impaired neutrophil function, collectively weakening antimicrobial defense (42). The risk associated with specific antibiotic classes reflects differences in resistance selection pressure; broad-spectrum cephalosporins, carbapenems, etc., are key drivers for MDROs like ESBLs and CRE, with prolonged use increasing MDRO detection rates in gut and sputum to over 60% (43, 44). Furthermore, the model effectively quantifies healthcare-associated risks through parameters including prolonged hospitalization and high SAPS II scores reflecting physiological derangement. By synthesizing these multidimensional signals—from host vulnerability to healthcare pressures—our model moves beyond traditional single-factor risk assessment, offering a unified mechanism to identify patients caught in a vicious cycle where systemic inflammation triggered by CRBSI predisposes them to MDRO infection.

This study employed logistic regression alongside seven ML algorithms to build prediction models, balancing "multivariable processing" and "interpretability," aligning with trends in severe infection prediction. Logistic regression, a traditional statistical method, offers strengths in interpretability, allowing clear quantification of each variable's risk contribution through odds ratios and 95% confidence intervals, making results easily understandable and acceptable to clinicians (45). In contrast, the seven ML algorithms primarily address limitations of traditional methods in parsing complex data, excelling particularly at capturing non-linear relationships and potential interactions among variables. Among them, gradient boosting tree algorithms performed exceptionally well: Cui et al. demonstrated in predicting MDRO in VAP that the XGBoost model, due to its robust handling of complex data interactions, significantly outperformed other algorithms (AUC = 0.831) (29). Similarly, other gradient boosting trees like LightGBM have shown excellent multi-feature processing capability and strong generalization performance (46). Furthermore, the built-in regularization mechanisms in gradient boosting trees effectively control model complexity and mitigate overfitting risk; their capacity for parallel computing significantly enhances training efficiency when handling large clinical data sets, while their flexibility with incomplete or heterogeneous clinical data can reduce prediction bias caused by data quality fluctuations (47). These characteristics align well with the practical needs of CRBSI prediction—high data dimensionality, significant sample heterogeneity, and the requirement for efficient, stable clinical decision-making—providing methodological support for our ML algorithm selection and achieving an organic unity of interpretability and complex data processing capability.

## Limitations

This study, based on the MIMIC-IV database and real-world multi-center data, developed and validated a ML model for predicting MDRO infection risk in CRBSI patients. Utilizing the mlr3 ML framework and standardized statistical methods, we obtained a robust prediction model and enhanced its clinical interpretability using SHAP. However, several limitations warrant mention. First, the retrospective study design carries inherent risks of bias. The use of database and multi-center retrospective data may introduce selection and measurement biases, potentially affecting the model's stability and generalizability. Specifically, limited by data completeness, we could only identify CRBSI patients in the MIMIC database using ICD diagnosis codes combined with matching identical pathogens, lacking access to key information like quantitative blood culture counts and time-to-positivity. This might lead to some false-positive CRBSI diagnoses and an inability to fully exclude other infection sources. Furthermore, certain variables with established importance for MDRO risk, such as the precise duration of antibiotic exposure prior to the CRBSI event, history of hospitalization within the preceding 90 days, and the use of other invasive devices, were not systematically recorded in the retrospective databases used and thus could not be included. Similarly, data on prior MDRO colonization or infection status, which typically requires proactive admission screening, were unavailable due to the lack of universal screening protocols across the participating centers. Additionally, data limitations include missing dynamic variables. This study incorporated only baseline or static variables, failing to capture the dynamic changes in indicators during disease progression. Given the often rapid progression of CRBSI, such dynamic parameters might better reflect the true risk of MDRO infection. Second, variations in data quality, sample size, population representativeness (48), and timeframes across the multi-center data may introduce inter-sample heterogeneity, affecting the model's consistency estimates. Third, despite our efforts to control for known confounders, we cannot fully rule out potential influences from unmeasured factors, such as bacterial colonization (16), blood culture contamination, or contamination of catheter dressings/fixation devices (49, 50), on both CRBSI diagnosis and model predictions. Finally, a translational shortcoming lies in the lack of integration of multi-modal data, posing obstacles for further clinical application and broad generaliza-

bility. Future research should focus on conducting multi-center prospective validation in diverse populations and refining the model to enhance its clinical translation value.

## Conclusion

In summary, addressing the challenges of delayed clinical identification of MDR-CRBSI and the limitations of traditional methods, this study employed a retrospective dual-cohort design. Using the MIMIC-IV database as the development set and data from three regional medical centers in China as external validation sets, it incorporated multi-dimensional variables including demographics, laboratory indices, and comorbidities. Following feature selection via LASSO regression, prediction models were constructed using logistic regression and seven ML algorithms, with interpretability enhanced by SHAP. Results demonstrated that the optimal model performed robustly in external validation, exhibited good generalizability, and identified core risk factors such as RDW, PLT, and antibiotic class. Although limitations exist, including biases inherent to the retrospective design and constrained population representativeness, this study provides a valuable predictive tool for the early identification of high-risk MDR-CRBSI patients and for informing precise treatment strategies.

## ACKNOWLEDGMENTS

The authors would like to thank the MIMIC database for providing data access and everyone who participated in the multicenter data collection.

This research was supported by the Science and Technology Innovation Project of Colleges and Universities in Shanxi Province (2024L090 and 2024SJ196), the Research project of High-quality Development in Shanxi Province (SXGZL202423), and the Research Project of Science and Technology Innovation Think Tank Construction of Shanxi Association for Science and Technology (KXKT202318).

H.W., M.Z., and C.Y. performed the data analyses, established the ML models, and drafted the manuscript. F.R. and X.W. participated in data collection, data analyses, and confirmed the validity of the experiments. H.Y., W.Q., F.T., and L.L. participated in the design of the study and coordination. H.W. had primary responsibility for study design, data analyses, data interpretation, and writing the manuscript. F.T., M.Z., and C.Y. have accessed and verified the data. All authors read and approved the final manuscript.

## AUTHOR AFFILIATIONS

[1]Department of Neurosurgery, Shanxi Provincial People's Hospital, The Fifth Clinical Medical College of Shanxi Medical University, Taiyuan, China
[2]School of Nursing, Shanxi Technology and Business University, Taiyuan, China
[3]Department of Infectious Diseases, Ningxia Medical University General Hospital, Yinchuan, Ningxia Hui Autonomous Region, China
[4]School of Clinical Simulation, Shanxi Medical University, Taiyuan, China
[5]Department of Medical Oncology, Peking Union Medical College Hospital, Chinese Academy of Medical Sciences, Beijing, China
[6]Department of Hospital-Acquired Infection Management, Second Hospital of Shanxi Medical University, Taiyuan, China
[7]Department of Hospital-Acquired Infection Management, Shanxi Provincial People's Hospital, The Fifth Clinical Medical College of Shanxi Medical University, Taiyuan, China

## AUTHOR ORCIDs

Hongwei Wang http://orcid.org/0009-0005-4294-1102
Caizheng Yang http://orcid.org/0009-0004-0249-7337
Fangying Tian http://orcid.org/0000-0003-3083-4287
Linping Li http://orcid.org/0009-0002-8664-9566

## AUTHOR CONTRIBUTIONS

Hongwei Wang, Conceptualization, Data curation, Formal analysis, Funding acquisition, Investigation, Methodology, Resources, Software, Validation, Writing – original draft, Writing – review and editing | Caizheng Yang, Conceptualization, Data curation, Formal analysis, Funding acquisition, Validation, Visualization, Writing – original draft | Ming Zhao, Conceptualization, Data curation, Investigation, Methodology, Resources, Writing – original draft | Fen Ren, Data curation, Investigation, Methodology, Project administration, Resources, Validation, Writing – original draft | Xueyu Wang, Conceptualization, Investigation, Project administration, Validation, Visualization, Writing – original draft | Haihua Yan, Conceptualization, Investigation, Methodology, Project administration, Resources, Supervision, Validation, Visualization, Writing – original draft | Weiwei Qin, Conceptualization, Investigation, Methodology, Project administration, Resources, Supervision, Validation, Visualization, Writing – original draft | Fangying Tian, Conceptualization, Data curation, Funding acquisition, Investigation, Methodology, Project administration, Supervision, Validation, Writing – original draft | Linping Li, Conceptualization, Formal analysis, Funding acquisition, Investigation, Methodology, Project administration, Resources, Supervision, Validation, Visualization, Writing – original draft, Writing – review and editing

## DATA AVAILABILITY

The data analyzed and the codes used during the current study are available from the corresponding author on reasonable request.

## ETHICS APPROVAL

The data collection protocol was reviewed and approved by the Ethics Committee of the SXMUSH (Approval No. 2024YX-159). The ethics committee waived the requirement for informed consent due to the retrospective nature of the study.

The design and reporting of this study strictly adhered to the Transparent Reporting of a multivariable prediction model for Individual Prognosis Or Diagnosis (TRIPOD) guidelines (51).

## ADDITIONAL FILES

The following material is available online.

### Supplemental Material

**Tables S1 to S4 (Spectrum03713-25-s0001.docx).** Test for multicollinearity, comparison of performance of each model on training and validation sets, and optimal parameter combination.

### Open Peer Review

**PEER REVIEW HISTORY (review-history.pdf).** An accounting of the reviewer comments and feedback.

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
