## [Reviewer comments · Microbiology Spectrum]

Microbiology Spectrum

Comparison and Validation of Multiple Machine Learning Algorithms for Predicting MDRO Infection in catheter-related bloodstream Patients: a multicenter cohort study

Hongwei Wang, Caizheng Yang, Ming Zhao, Fen Ren, Xueyu Wang, Haihua Yan, Weiwei Qin, Fangying Tian, and Linping Li

Corresponding Author(s): Hongwei Wang, Fifth Hospital of Shanxi Medical University

Review Timeline:

Submission Date:	November 16, 2025
Editorial Decision:	December 22, 2025
Revision Received:	December 29, 2025
Accepted:	January 6, 2026

Editor: Gabriele Arcari

Reviewer(s): Disclosure of reviewer identity is with reference to reviewer comments included in decision letter(s). The following individuals involved in review of your submission have agreed to reveal their identity: AHMAD AHMAD (Reviewer #1)

Transaction Report:

DOI: <https://doi.org/10.1128/spectrum.03713-25>

Re: Spectrum03713-25 (**Comparison and Validation of Multiple Machine Learning Algorithms for Predicting MDRO Infection in catheter-related bloodstream Patients: a multicenter cohort study**)

Dear Prof. Linping Li:

Thank you for the privilege of reviewing your work. Below you will find my comments, instructions from the Spectrum editorial office, and the reviewer comments.

Revision Guidelines

Sincerely,
Gabriele Arcari
Editor
Microbiology Spectrum

Reviewer #1 (Comments for the Author):

In their manuscript titled "Comparison and Validation of Multiple Machine Learning Algorithms for Predicting MDRO Infection in catheter-related bloodstream Patients: a multicenter cohort study" the authors have trained eight machine learning models to predict patients with MDRO-CRBSI risk using clinical data. The experiments of the study are well planned, and manuscript is very well written. I have the following few minor comments.

1. The main concern is the small number of sample size of the study. Generally, machine learning (ML) models need large number of samples size.
2. Line 235: How much samples was generated using SMOTE, please mention. Also, was the generated samples subjected to quality checks?
3. Line 237: Feature selection. It will be better to mention starting feature size and the final feature size.
4. Line 254: Was the study finally performed on 9 variables or 43 variables, please mention clearly.
5. Line 296- 298: It seems the hospital length of stay is main factor for MDR-CRBSI, how many patients with 5 or 6 days stay got MDR-CRBSI?
6. Figure 2 a,b,e,f,g: and Line 314 -325: Are these results of a single model or 8 models? It seems that a single model was used to tune max_depth, learning rate, number of tuning attempts etc. Which algorithm was used for this tuning. Logistic regression, XGboost etc. As all algorithms have their own hyperparameters to tune. How were they tuned again?
7. Line 32: It should read as "Patient data with extracted"

Reviewer #2 (Comments for the Author):

Very interesting article. Few comments:

1. Many acronyms related to ML have been used, since it is going to be read by medicos, it would be good to give the full forms at least once.
2. Line no. 96 mentions: WBC count demonstrates markedly different discriminatory power between gram positive and gram negative: Could you please elaborate on this.
3. Apart from the factors analysed, certain important points which determine the increased likelihood of MDROs include history of hospitalisation in the last 90 days, any other invasive devices.
3. Duration of exposure to antibiotics, history of MDROs in the recent past should have been considered.
4. In your discussion, 405-406, malignancies induce immunosuppression; however additionally recurrent admissions and exposure to healthcare facilities are also responsible for increased risk of MDROs.

In their manuscript titled “**Comparison and Validation of Multiple Machine Learning Algorithms for Predicting MDRO Infection in catheter-related bloodstream Patients: a multicenter cohort study**” the authors have trained eight machine learning models to predict patients with MDRO-CRBSI risk using clinical data. The experiments of the study are well planned, and manuscript is very well written. I have the following few minor comments.

1. The main concern is the small number of sample size of the study. Generally, machine learning (ML) models need large number of samples size.
2. Line 235: How much samples was generated using SMOTE, please mention. Also, was the generated samples subjected to quality checks?
3. Line 237: Feature selection. It will be better to mention starting feature size and the final feature size.
4. Line 254: Was the study finally performed on 9 variables or 43 variables, please mention clearly.
5. Line 296- 298: It seems the hospital length of stay is main factor for MDR-CRBSI, how many patients with 5 or 6 days stay got MDR-CRBSI?
6. Figure 2 a,b,e,f,g: and Line 314 -325: Are these results of a single model or 8 models? It seems that a single model was used to tune max_depth, learning rate, number of tuning attempts etc. Which algorithm was used for this tuning. Logistic regression, XGboost etc. As all algorithms have their own hyperparameters to tune. How were they tuned again?
7. Line 32: It should read as “Patient **data** with extracted”

Response to Reviewers

(Manuscript ID: Spectrum03713-25)

1. Detailed responses to Reviewer 1's comments

Reviewer's Comment 1: The main concern is the small number of sample size of the study. Generally, machine learning (ML) models need large number of samples size.

Author's reply: We thank the reviewers for raising this important point. We acknowledge that, for machine learning, a larger sample size is generally preferable. However, due to the relatively low incidence of catheter-related bloodstream infections (CRBSI) in the population and the even lower incidence of cases caused by multidrug-resistant bacteria (MDRO), there are objective epidemiological limitations in obtaining a large sample. This reality is also one of the reasons why this clinical prediction problem is challenging.

In this context, although the sample size of the development cohort in this study is limited, we have targeted the adoption of multiple methodological strategies to maximize the robustness and reliability of the model:

1. Strict feature selection: We used LASSO regression to select 10 core variables from 51 initial variables, significantly reducing the risk of overfitting in high-dimensional data.
2. Repeated cross-validation: The training and hyperparameter optimization of all models were conducted through repeated (50 iterations) 5-fold cross-validation, thereby obtaining more stable and reliable performance estimates within the limited data.
3. Independent external validation: The most critical evaluation of the model lies in its performance in a new, multi-center external validation cohort (n=362). The model maintained a high discriminatory efficacy, which strongly demonstrated its good generalization ability and effectively

alleviated concerns about overfitting in the development set.

4. Handling class imbalance: We applied the SMOTE technique to the training set to correct the imbalance in the number of MDRO and non-MDRO cases.

In conclusion, although a larger development cohort would theoretically have advantages, considering the epidemiological characteristics of the disease and the above methodological design, the model constructed in this study has demonstrated superior efficacy and stability. Moreover, we have supplemented this point in the “Limitations” section of the paper, and will further expand the sample size through multi-center collaboration in the future.

Reviewer's Comment 2: Line 235: How much samples was generated using SMOTE, please mention.

Also, was the generated samples subjected to quality checks?

Author's reply: We thank the reviewer for raising these important methodological points. Number of synthetic samples: As now detailed in the revised “Data Processing and Quality Control” section, SMOTE was applied to the training cohort which initially contained 189 MDR-CRBSI (minority class) and 1,062 non-MDR-CRBSI (majority class) patients. A total of 873 synthetic samples were generated for the minority class, creating a balanced training set of 1,062 samples per class (total $n=2,124$). Validation of the oversampling approach: The “quality” or utility of SMOTE-generated samples is optimally assessed by the performance of the final model trained on the balanced dataset. In our study, the model trained on the SMOTE-balanced data achieved high performance for the minority class (recall/sensitivity = 0.704) while maintaining excellent overall discriminative ability (AUC=0.877) and a very low false-positive rate in the training set. Crucially, this robust performance generalized well to the independent external validation set (AUC=0.851). This consistency confirms that the oversampling effectively improved the model's learning of the minority class characteristics

without introducing overfitting to artificial patterns.

Reviewer's Comment 3: Line 237: Feature selection. It will be better to mention starting feature size and the final feature size.

Author's reply: We thank the reviewer for this constructive suggestion. To enhance clarity, we have revised the "Feature Selection" section to explicitly state the progression of variable numbers. The modified text now reads: "Starting from an initial pool of 51 candidate variables, which was refined to 43 variables after data preprocessing. This refined procedure resulted in the retention of 10 variables with non-zero coefficients for final model development." This addition provides a clear documentation of our feature selection pipeline.

Reviewer's Comment 4: Line 254: Was the study finally performed on 9 variables or 43 variables, please mention clearly.

Author's reply: We sincerely thank the reviewer for this important question. The final prediction models in our study were developed and validated using 10 key predictive features (e.g., RDW, CRP, PLT) selected by the elastic net model.

To trace the full process:

The analysis started with an initial pool of 51 candidate variables.

After data preprocessing and the removal of redundant variables (detailed in the Data Processing and Quality Control section), 43 variables were retained for the feature selection stage.

The sequential feature selection process (LASSO followed by elastic net) distilled these 43 variables down to the final set of 10 variables with non-zero coefficients.

To prevent any ambiguity, we have made the following clarifying revisions to the manuscript:

In the "Feature Selection" section, the text now explicitly states: "Starting from an initial pool of

51 candidate variables... this refined procedure resulted in the retention of 10 variables with non-zero coefficients for final model development.”

In the “Model Development and Parameter Optimization” section, we have added: “All subsequent models were developed using the final set of 10 predictor variables selected through the feature selection process described above.”

We hope these revisions in the manuscript, alongside this explanation, fully address the reviewer's query regarding the number of variables used in our analysis.

Reviewer's Comment 5: Line 296- 298: It seems the hospital length of stay is main factor for MDR-CRBSI, how many patients with 5 or 6 days stay got MDR-CRBSI?

Author's reply: We thank the reviewer for this insightful question, which allows us to examine the relationship between length of stay (LOS) and MDR-CRBSI risk in greater clinical detail. Upon additional descriptive analysis of our training cohort, we identified all patients with a hospital stay of 5 or 6 days (inclusive):

Total number of patients with LOS 5-6 days: 141

Number of these patients diagnosed with MDR-CRBSI: 13

Proportion with MDR-CRBSI in this LOS window: 9.2% (13/141)

This finding aligns with the reviewer's intuition that LOS alone, at moderate durations, is not a predominant determinant of infection. The vast majority (128 out of 141) of patients hospitalized for 5-6 days did not develop MDR-CRBSI. This observation underscores the value of a multivariable approach. In our XGBoost model, LOS is not a standalone linear predictor but one integrated feature among ten. The model likely captures a non-linear or interactive effect, where the risk associated with a given LOS is substantially modulated by the values of other features (e.g., inflammation

markers, comorbidities). The model's power lies in identifying the high-risk minority even within this ostensibly lower-risk LOS stratum by synthesizing the complete clinical picture. Its predictive contribution is nuanced and contextual, dependent on its interplay with other variables. Thank you for prompting this valuable clarification, which strengthens the interpretation of our model's predictors.

Reviewer's Comment 6: Figure 2 a,b,e,f,g: and Line 314 -325: Are these results of a single model or 8 models? It seems that a single model was used to tune max_depth, learning rate, number of tuning attempts etc. Which algorithm was used for this tuning. Logistic regression, XGboost etc. As all algorithms have their own hyperparameters to tune. How were they tuned again?

Author's reply: We sincerely thank the reviewer for this critical observation, which highlights an area where our description could have been clearer. We apologize for any confusion. The reviewer is absolutely correct that each of the eight models has its own distinct hyperparameters and requires independent tuning. Our methodology indeed involved conducting separate hyperparameter optimization for each model (logistic regression and the seven machine learning algorithms).

To clarify:

Content of Figures 2e-g: These figures specifically illustrate the tuning process for the XGBoost model only. We chose to visualize XGBoost's tuning in the main text because it was ultimately selected as the best-performing model.

Documentation of All Results: The outcome of this individualized tuning process for all eight models is fully documented in Supplementary Table 4 (provided in our initial submission), which lists the unique optimal hyperparameters and the corresponding cross-validated AUC for each algorithm.

In direct response to the reviewer's questions:

Which algorithm's tuning is shown in Fig 2e-g? XGBoost.

How were the others tuned? Each underwent the same 50-iteration random search cross-validation procedure. The optimal parameters for Logistic Regression, RF, SVM, etc., are found in Supplementary Table 4.

Reviewer's Comment 7: Line 32: It should read as "Patient data with extracted"

Author's reply: Thank you for pointing this out. The sentence has been corrected as per your suggestion.

2. Detailed responses to Reviewer 2's comments

Reviewer's Comment 1: Many acronyms related to ML have been used, since it is going to be read by medicos, it would be good to give the full forms at least once.

Author's reply: We thank the reviewer for this important suggestion, which has significantly improved the readability and clarity of the manuscript. We have ensured that all nonstandard abbreviations are defined upon their first use in the text. Definitions have been added for the following terms: LASSO (Least Absolute Shrinkage and Selection Operator) and VIF (variance inflation factor).

Reviewer's Comment 2: Line no. 96 mentions: WBC count demonstrates markedly different discriminatory power between gram positive and gram negative: Could you please elaborate on this.

Author's reply: We thank the reviewer for this observation. We agree that the original phrasing was potentially misleading. Our intended point, supported by the cited literature, is that the predictive value of individual biomarkers like white blood cell count is not only limited but also highly variable and inconsistent across different types of infections (e.g., Gram-positive vs. Gram-negative), thus

making it an unreliable standalone predictor. We have revised the sentence in the manuscript to clarify this meaning.

Reviewer's Comment 3: Apart from the factors analysed, certain important points which determine the increased likelihood of MDROs include history of hospitalisation in the last 90 days, any other invasive devices.

Author's reply: We sincerely thank the reviewer for raising these crucial clinical points. The reviewer is absolutely correct that detailed longitudinal exposure histories, such as the duration of recent antibiotic therapy and prior MDRO colonization status, are critical for MDRO risk assessment. We fully acknowledge this limitation in our study. Our model was constrained by the variables consistently and systematically available in the retrospective databases we utilized (MIMIC-IV and the multi-center hospital databases). Specifically:

Duration of Antibiotic Exposure: The precise duration of antibiotic treatment prior to the CRBSI event was not systematically recorded in these databases.

Prior MDRO Colonization/Infection Status: Obtaining this information reliably would require proactive universal screening upon patient admission, which was not performed in the included centers due to resource and protocol constraints. Therefore, this data was unavailable for our cohort.

We have now clearly stated this point in the revised "Limitations" section: "Furthermore, certain variables with established importance for MDRO risk, such as the precise duration of antibiotic exposure prior to the CRBSI event, history of hospitalization within the preceding 90 days, and the use of other invasive devices, were not systematically recorded in the retrospective databases used and thus could not be included. Similarly, data on prior MDRO colonization or infection status, which typically requires proactive admission screening, were unavailable due to the lack of universal

screening protocols across the participating centers.”

We appreciate the reviewer’s insightful comment, which has helped us improve the transparency and discussion of our work.

Reviewer's Comment 4: Duration of exposure to antibiotics, history of MDROs in the recent past should have been considered.

Author's reply: We appreciate the reviewer's comment, which highlights the critical importance of incorporating detailed antibiotic exposure history and prior MDRO status into risk assessment models for MDR-CRBSI. These points are indeed highly relevant and are closely related to the broader data limitations discussed in our response to Comment #3 above.

As detailed in our reply to Comment #3, our retrospective, database-driven study design imposed practical constraints that prevented the inclusion of these specific variables. The precise duration of antibiotic exposure prior to the CRBSI event was not systematically recorded in the MIMIC-IV or our multi-center hospital databases. Similarly, reliable data on prior MDRO colonization or infection history was unavailable, as its ascertainment would require proactive, universal admission screening—a protocol not in place across the participating centers.

We acknowledge that the absence of these variables represents a limitation that may affect the model's comprehensiveness. To ensure transparency, we have incorporated a clear statement addressing this point in the revised "Limitations" section of the manuscript. We agree with the reviewer that future prospective studies, specifically designed to capture these detailed exposure histories, would be invaluable for further refining the predictive accuracy and clinical utility of models for MDR-CRBSI. We thank the reviewer for underscoring these important considerations.

Reviewer's Comment 5: In your discussion,, 405-406, malignancies induce immunosuppression;

however additionally recurrent admissions and exposure to healthcare facilities are also responsible for increased risk of MDROs.

Author's reply: We thank the reviewer for this valuable suggestion to broaden the clinical perspective of our discussion. We fully agree that recurrent healthcare exposure is a critical, independent risk factor for MDRO acquisition. As suggested, we have expanded the relevant paragraph in the Discussion section to explicitly incorporate this point.

Re: Spectrum03713-25R1 (**Comparison and Validation of Multiple Machine Learning Algorithms for Predicting MDRO Infection in catheter-related bloodstream Patients: a multicenter cohort study**)

Dear Dr. Hongwei Wang:

Your manuscript has been accepted, and I am forwarding it to the ASM production staff for publication. Your paper will first be checked to make sure all elements meet the technical requirements. ASM staff will contact you if anything needs to be revised before copyediting and production can begin. Otherwise, you will be notified when your proofs are ready to be viewed.

Sincerely,
Gabriele Arcari
Editor
Microbiology Spectrum